# 3D Reality-Based Survey and Retopology for Structural Analysis of Cultural Heritage

**DOI:** 10.3390/s22249593

**Published:** 2022-12-07

**Authors:** Sara Gonizzi Barsanti, Mario Guagliano, Adriana Rossi

**Affiliations:** 1Department of Engineering, Università deli Studi della Campania Luigi Vanvitelli, 81031 Aversa, Italy; 2Department of Mechanical Engineering, Politecnico di Milano, 20156 Milan, Italy

**Keywords:** 3D modelling, 3D survey, retopology, NURBS, FEA, convergence analysis

## Abstract

Cultural heritage’s structural changes and damages can influence the mechanical behaviour of artefacts and buildings. The use of finite element methods (FEM) for mechanical analysis is largely used in modelling stress behaviour. The workflow involves the use of CAD 3D models and the use of non-uniform rational B-spline (NURBS) surfaces. For cultural heritage objects, altered by the time elapsed since their creation, the representation created with the CAD model may introduce an extreme level of approximation, leading to wrong simulation results. The focus of this work is to present an alternative method intending to generate the most accurate 3D representation of a real artefact from highly accurate 3D reality-based models, simplifying the original models to make them suitable for finite element analysis (FEA) software. The approach proposed, and tested on three different case studies, was based on the intelligent use of retopology procedures to create a simplified model to be converted to a mathematical one made by NURBS surfaces, which is also suitable for being processed by volumetric meshes typically embedded in standard FEM packages. This allowed us to obtain FEA results that were closer to the actual mechanical behaviour of the analysed heritage asset.

## 1. Introduction

Accurate 3D reality-based documentation is a must have for proper preservation and conservation in the cultural heritage field, as it is a prerequisite for more analyses. This documentation and these analyses are of more importance in contemporary times because of atmospheric agents, the growing of the cities and of the density of constructions, carelessness over the centuries, and the present political instability in certain areas that have all affected and strongly influenced the solidity of our heritage. It is hence fundamental to complete diagnostic studies aimed at valuing the level of decay of cultural heritage for selecting the appropriate preservation methods. However, it is challenging to calculate how a historical artefact suffers for environmental agents (e.g., earthquakes, pollution, wind, and rain) or human factors (e.g., construction in the environments, vehicular traffic, dense tourism). Hence, it is mandatory to find the best pipeline to obtain results as close as possible to reality. Finite element analysis (FEA) is a recognised technique used in engineering for various purposes (e.g., for modelling stress behaviour under mechanical and thermal loads), starting from a CAD 3D model made by non-uniform rational B-spline (NURBS) surfaces. Once imported in an FEA software, these 3D closed models can be meshed using modules capable of transforming a surface model to a volumetric one, which has nodes allocated in both the exterior and the interior volume, joined by simple volumes such as tetrahedron, pyramids, prisms, or hexahedral. In the mechanical area, this workflow is applicable because the 3D digital object to be simulated is close to the reality, within strict tolerances. Contrarywise, when applied to 3D models of cultural heritage (CH) objects or structures, the representation with a CAD model introduces a disproportionate level of approximation that can lead to incorrect simulation outcomes. Today, the 3D documentation of CH has been extensively matured through active sensors or passive approaches such as photogrammetry. The model obtained through these techniques is a surface formed by millions of triangles and is not suitable for direct use in FEA because the software is not able to manage so many polygons, and the computational complexity of FEA grows exponentially with the number of nodes representing the simulated object. Hence, a simplification is needed, and then a transformation of the superficial 3D meshes in volumetric models is needed to be meshed in the FEA software accordingly. Preliminary experiments were carried out on real CH artefacts surveyed with active or passive methods [1,2] for simulating stress behaviour and predicting critical damages. Analysing the results, few issues were made evident: (a) the creation of a volumetric model to be used in the FEA software from the raw 3D data is not yet clearly defined and may greatly affect the result, (b) the balance between geometric resolution and the accuracy and precision level of the simulated results is often not compatible with the shape of a 3D reality-based model.

The approaches used to generate the volumetric model from the acquired 3D point cloud are different: (a) using CAD software for the drawing of a new surface model following the superficial mesh originated by the acquired 3D cloud [3]; (b) using the triangular mesh generated by the 3D survey [4]; (c) generating a volumetric model from the 3D point cloud without preliminary surface meshing; (d) using the 3D reality-based model as the basis for a BIM/HBIM for FEA [5,6,7]; (e) creating new tools (Figure 1).

The first methodology was used, for example, to simulate the structural behaviour of the Trajan’s Markets [3] and in many other applications [8,9,10,11]. In some cases, the mathematical model was improved with the insertion of patches of reality-based meshes [12,13] that also used a strong discretization of the model using both a 3D CAD model and a 3D composite beam one [14,15]. Some projects used a joined system to obtain the 3D model, starting from the use of reality-based techniques, GPR, and radar [15]. Extracting or drawing cross-sections and profiles from the 3D reality-based model are ways to produce the CAD model extrapolated from a reality-based one, as in [16,17,18]. This procedure has its limitation related to the re-draw of the model in a CAD environment, but it can be applied to CH buildings for which the structure and the geometric details can be replicated through a CAD drawing using profiles. On the contrary, it cannot be used for statues, whose geometry is more complex and cannot be reduced through elements such as beam, truss, or shell, which are used for modelling in FEA.

The second approach has a variety of different methods: (i) plain simplification of the triangular mesh before converting it to a volumetric one, which may have important differences between the real shape and the simulated one [19]; (ii) simplified depiction of the shape as discretized profiles offering a low-resolution representation of the interior and the exterior of the structure, from which generate a volumetric model can be created [5,7]; the fitting of the acquired 3D model with a parametric one suitable to be transformed to volume [20].

The use of the triangular mesh for the structural analysis to assess the stability of a marble statue is highlighted in [21], analysing the mechanical stresses generated on the statue and the pedestal materials. The procedure begins with the subtraction of the mesh from a prismatic block shape and is called FEA in situ (the algorithm performs the analysis directly on the mesh without passing through the volumetric model). Some tests on mechanical objects were presented to corroborate this method.

The third strategy does not even take into consideration the mesh because it creates a volumetric approximation of the shape of the original 3D model from the raw 3D cloud of points [22], later compared by the same authors to other approaches [6,23]. The fourth methodology uses a 3D reality-based model to produce HBIM models to be used in FEA, implying two levels of approximation: (i) the first one related to the drawing of a BIM model from 3D reality-based mesh; (ii) creating the volumes for FEA starting from the BIM models. In [24], the authors investigated the use of BIM models to assist sites with monitoring and management, extrapolating thematic information for structural analysis, even if the authors did not provide a finite element analysis on the structure. Another example is the creation of the HBIM from both archival data and a laser scan survey. The model created was then segmented and utilised for structural analysis [25]. The Masegra Castle, located near the city of Sondrio in Italy, served as a test object for an original procedure called Cloud-to-BIM-to-FEM [26]. In this case, the basis for the HBIM was an accurate survey combining geometrical features, diagnostic analysis based on destructive and non-destructive inspections, material data, element interconnections, and architectural and structural considerations. This model was then converted into a finite element model with a geometric rationalization, considering irregularities and anomalies (e.g., verticality deviation and variable thickness).

Directly using the 3D reality-based models in FEA has its advantages because it overcomes all the approximations seen in the previous works. It is necessary to simplify the 3D meshes originated by this type of survey to make them suitable for FEA software. The best procedure for this, maintaining the accuracy of the 3D reality-based model, is using retopology, which implies a strong simplification of the mesh connected to the topological rearrangement of it, hence the creation of a new topology for the 3D model [27,28]. The retopologised mesh is normally based on quadrangular element (quads). Its main advantage is the reorganization of the polygonal superficial elements of the meshes for their better distribution on the surface. This procedure allows for the application of a huge simplification of the meshes without losing the initial accuracy of the models. The more organised structure of the elements on the mesh helps also in the conversion of it in NURBS, while maintaining a better coherence with the digitized artefact. This is valuable when working with reality-based 3D models of cultural heritage, usually accurate and precise but with a complex geometry.

The method proposed is based on the simplification of 3D meshes and their export into mathematical ones, close as much as possible to the real shape of the object surveyed but suitable to be converted into rationally complex volumetric 3D models. This permits us to obtain volumetric models of cultural heritage artefacts, increasing the closeness of the resultant NURBS model with the acquired one and, in the meantime, reducing the number of NURBS patches necessary for describing it. This methodology can be useful for experts such as archaeologists, architects, restores, and structural engineers, given that the lack of funding usually affects the restoration’s interventions. An accurate pipeline can help in locating probable causes for future problems, allowing a more effective conservation of the artefact.

Three different case studies are discussed, showing the accuracy of the methods and the application on real cultural heritage objects.

## 2. Materials and Methods

The most complex part of executing structural analysis on cultural heritage artefacts is related to the geometrical complexity of the object analysed and the fact that they, especially buildings, are built with different construction techniques and different materials. This circumstance leads to the consideration of different points when dealing with structural analysis in the field of cultural heritage. The intrinsic characteristics of the different elements involved in the pipeline and the non-linear and non-symmetric geometry of the structures influence the choice of the procedure. Therefore, the methodology proposed took into consideration the use of retopology for the decimation of the reality-based 3D models.

## 3. 3D Modelling, Post-Processing, and Orientation of the Model

The test objects were surveyed with photogrammetry and laser scanning, considering the final aim of the modelling. The GSD (ground sample distance) and the accuracy of the scanner used were taken into consideration to obtain precise and accurate models and to obtain a value to compare the following stages of the pipeline proposed. For the validation of the methodology, a lab steel specimen for mechanical testing and a violin were used. Then, the methodology was applied to a statue of the Uffizi Museum of Florence (Figure 2a–c). The differences in these test objects were that, for the lab specimen, the analytical results of the mechanical tests were known, and for the violin, the results of FEA were compared to direct tests in the laboratory. Once the procedure was tested and validated, it was applied to a statue for which no analytical comparison was available.

The lab specimen has a cylindrical shape with a groove and is 148.28 mm long, with the larger diameter of 20 mm and the smaller of 7.4 mm. The choice of this object was made considering (i) its shape that originated from the revolution of a profile defined by an analytical function. This allowed us to calculate the stresses in critical points relatively easily, thus permitting the assessment of the results of an FE run with respect to the analytical solution. The results obtained can be used as theoretical reference; (ii) the physical object can be used for laboratory tests in different stress conditions, experimentally measuring its mechanical behaviour, which can be used as an experimental reference; (iii) the 3D model of the object can be generated with 3D digitization, and the FEA can be applied to different instances of the 3D model created with different 3D simplification methods, allowing us to prove the value of the proposed method. The specimen was surveyed with a structured light Solutionix Rexcan CS device (Table 1).

The blue-light sensor for the pattern projection is suitable for digitizing small and medium not-totally Lambertian objects and is considered the most precise type of sensor for 3D digitization in mechanical engineering. The survey was carried out by locating the optical head on a base connected to a rotating plate (TA-300) composed of two axes, one for rotation around the vertical direction and one for the oscillation. The rotation allows for movement of ± 180°, and the axis of oscillation allows the inclination of the plane, where the specimen is located, up to 45° with respect to the vertical direction. With this scanning range, it is possible to limit to the minimum blind spots by reducing the scanning time up to 40%. The range device can mount three different lenses, for which the specifications are summarized in Table 2. Given the size of the test object, wide-angle lenses (12 mm) that were calibrated using the appropriate calibration table fixed on the turntable were chosen.

The 3D device works with proprietary software for the acquisition and alignment phases. For the survey, the Multiscan setting was used with an oscillation of ±30° and ±150° rotations for a total of 36 scans for each position of the object on the turntable (12 scans, one for each rotation for the three different oscillations, −30°, 0°, and 30°). For the survey of the steel specimen, two groups of 36 scans were performed, automatically aligned by the Solutionix software, with a final RMS error of 18 μm.

The contemporary violin was surveyed with a six-axis arm laser scanner with a tolerance of 0.03 mm by the company that provided the mesh model. The violin was used because it was possible to conduct some acoustic test in the laboratory on the two separate sides of the artefact, which were then compared to the FEA on the retopologised models.

The statue was surveyed through photogrammetry using a Mirrorless APS-C SONY ILCE 6000 camera coupled with a 16 mm lens, which acquired the images even when the distance between the object and the camera was short. The distortion parameters of the lenses were corrected through the automatic calibration of the camera and the lens during the alignment phase in Agisoft Metashape, the software used to create the 3D model. The setting of the camera was ISO equal to 1000 and the focal length at 5.6. The GSD obtained was 1 mm (The survey was performed by Dott. Umair Shafquat Malik of Politecnico di Milano during a joint project with the Indiana University (coordinator Prof. Bernie Frisher). In 2019, the Uffizi Gallery in Florence with an agreement with Indiana University (IU-USA), started 3D digitization of its complete Roman and Greek sculptural collection in which the Reverse Engineering and Computer Vision group of Politecnico di Milano was involved as a technical partner, under the coordination of Prof. Gabriele Guidi [Virtual World Heritage Lab 2019].).

After the survey, the models were post-processed with the correction of the topological errors and of their orientation on a suitable reference system, with the XY plane corresponding to the base of the model and the *z*-axis passing from the centre of gravity of the artefact. The post-processed meshes were then simplified with retopology, converted in closed NURBS and then into volumetric models. The use of retopology is also valuable when converting the superficial meshes in NURBS because the process tends to generate a higher number of patches when the original mesh is topologically unorganized. Hence, the rearrangement of the initial topology of the mesh can be seen as a preliminary condition for minimizing the number of NURBS patches of the converted model. In this way, the mesh embedded in FEA software works better. After all these passages, the 3D models were finally prepared for the finite element analysis.

### 3.1. Simplification of the Models: Triangular and Retopology Method

3D models can be simplified with different strategies. The first approach is based on triangular simplification, and the second one involves the transformation of the original triangular mesh into a quadrangular one, its retopology and projection of the nodes on the original triangular mesh (Figure 3), according to the method described in [29]. The geometrical complexity of the models was exemplified in terms of nodes of the mesh (or vertexes) rather than in terms of polygons because the counting of polygons on a mesh is different if the shape is triangular or quadrangular. For the latter, the number of nodes and polygons is approximately the same, while for triangular meshes the number of polygons is approximately double that of the nodes (This is obvious since a squared element, once divided in two parts on one of its diagonals, produce two triangles.). It must be borne in mind that the number of vertexes of a mesh is what defines the surface sampling; therefore, it was used as an indicator for the level of detail of each mesh, independently of its triangular or quadrangular arrangement.

#### 3.1.1. Triangular Simplification

The triangular mesh is a set of vertexes V = (v1, v2, …, vk) and faces F = (f1, f2, …, fn). The simplification process obtains a surface M’ as similar as possible to the initial high resolution mesh M by lessening the number of the element on the surface. The simplification process is usually controlled by a set of user-defined quality criteria that can preserve specific properties of the original mesh as much as possible (e.g., geometric distance, visual appearance, etc.).

There are different approaches, the majority of which involves the degradation of the mesh to reduce the number of polygons [30,31]:Vertex decimation: it iteratively removes vertices and the adjacent faces. It preserves the mesh topology. The sequential optimization process manages the removal of points from the triangulation, leading to a gradual increase of its overall approximation error [32].Energy function optimization: the algorithm assigns an energy function to the number of nodes, the approximation error, and the length of the edges that regulate the regularity of the mesh. It produces higher results, minimizing the energy and solving the mesh optimization problems but increasing the computational cost.The agglomeration of vertices or vertex clustering: it partitions the mesh vertices into clusters and merges all the vertices in a cluster into one single vertex.Region merging–face clustering: it works on coplanarity. As a planarity threshold is set, the neighbourhood of each triangle is evaluated, and all the triangles that are inside the threshold are merged.

In this work, the first algorithm, implemented in the Polyworks software package (IMCompress), was used.

#### 3.1.2. Retopology

For the retopology process, this feature is available both in open-source packages, such as InstantMeshes [33] or Blender, or in commercial software packages, such as ZBrush by Pixologic, used in this work because this software is built specifically for rebuilding the topology of the models. Blender showed some problems in managing big files, while InstantMeshes, even if powerful, gave sometimes inadequate meshes as results, with holes and missing parts, especially when strongly decreasing the number of nodes. Moreover, ZBrush has the option of projecting the retopologised model onto the high resolution one, increasing the adherence of the two models.

The tool used for retopology was the ZRemesher, which is optimized to work on all kinds of structures and shapes but will by default produce better results with organic shapes, since Zbrush is specifically for the creation of video game characters. In the ZRemesher palette, there is the possibility to select the number of polygons desired for the retopology and the choice of increasing the coherence of the two models. This can be performed by selecting the “adapt” button and increasing the value of the “adapt size” slider. This is an important parameter for ZRemesher because it defines the polygon distribution on the model, and it can drastically increase the quality of the topology by giving more flexibility to the algorithm (Figure 4a,b).

This function defines a vertex ratio based on the curvature of the mesh. A low setting provides polygons that are as square as possible and almost the same size, a number of final polygons closer to the number set in the selection tool, but it can introduce topology irregularities where the geometry is more complex. A high adaptive size means obtaining polygons that are rectangular in shape to best fit the mesh’s curvature and for which density can vary along the mesh surface even if the program creates smaller polygons where the geometry requires. With a higher value of this parameter, there is less control on the final number of desired polygons after retopology. The adaptive size quantity goes from 0 to 100; it is not a unit, but it is a number that is only referred to for the different quantities and different settings of the quadrangular elements on the retopologised mesh.

Thus, by increasing the value of the adaptive size, the quality of the retopology increases but the program is more elastic regarding the target number of final polygons. This happens because, when the desired number of polygons is set, the software distributes them equally on the surface and then analyses the curvature, deforming the shape of the polygons or changing their density to be more adherent to the initial mesh.

### 3.2. NURBS Conversion

A mesh represents 3D surfaces with a series of discreet faces, similar to how pixels form an image. NURBS, on the contrary, are mathematical surfaces that can represent complex shapes with no granularity that is in mesh. The conversion from mesh to NURBS is implemented in CAD software or similar software (e.g., 3DMax, Blender, Rhinoceros, Maya, Grasshopper, etc.), and it transforms a mesh composed by polygons or faces to a faceted NURBS surface. In detail, it creates one NURBS surface for each face of the mesh and then merges everything into a single polysurface.

Depending on the mesh, the conversion works in different ways:If the starting point is a triangular mesh, and while, by definition, triangles are plane, the conversion creates trimmed or untrimmed planar patches. The degree of the patches is a 1 × 1 surface trimmed in the middle to form a triangle.If the starting point is a quadrangular mesh, the conversion creates four-sided untrimmed degree 1 NURBS patches, meaning that the edges of the mesh are the same as the outer boundaries of the patches.Considering the theory, a quadrangular mesh is more suitable to be converted into NURBS (Figure 5a,b). For this work, the MeshToNurb tool implemented in Rhinoceros was used.

### 3.3. Finite Element Analysis

The analysis was carried out on the models meshed using given elements, and they were different if a 2D or a 3D problem was evaluated. The 2D elements are the triangular and the quadrangular, while the 3D elements are the tetrahedral and the hexahedral; the hexahedral ones are more accurate (e.g., deform in a lower strain energy state) but it is more difficult to mesh a 3D volume with this kind of element if it is not segmented [34]. The 3D elements can be linear or quadratic; the difference is that the quadratic ones have nodes also on the mid side, varying in number from four nodes (linear tetrahedron) to 20 nodes (quadratic hexahedron), and the shape functions vary from linear to quadratic, allowing a more accurate description of the geometry and the displacement of the nodes. In the present study, the linear elements were chosen to simplify FE modelling from the geometrical model and to consider the convergence of the results by increasing the mesh density to assess the accuracy of the results.

## 4. Results

To validate the proposed method, some tests were conducted both on a case with a known analytical solution, using this one as a term of comparison, and on a case with experimental measurements conducted, also in this case to use the latter as a term of comparison to check the goodness of the developed finite element model and to check if the mesh size is adequate to obtain reliable results.

In fact, the best way to assess if a finite element model has the proper mesh size is to perform convergence studies by increasing the element count in the model and assuring that the result of interest graphically converges to a stable value. Mesh convergence in finite element analysis is related to the smallest dimension of the element of the mesh (how many elements are required in a model) to ensure that the results of the analysis are not influenced by the changing size of the mesh. At least three element dimensions are required to compute a convergence test, which happens when an asymptotic behaviour of the solution shows up, meaning that the difference among the results becomes smaller or equal. To determine mesh convergence, a curve of a stress parameter is required, plotted against different sizing in mesh density. The laboratory steel specimen was chosen as the test object because of its simple geometry and because it is possible to calculate the analytical result for the stress analysis; a simple traction analysis was carried out. Three different simplifications were used, triangular, retopology with the adaptive size parameter of 30 (low value), and retopology with the adaptive size parameter of 100 (highest value). It was decided to opt for these values to compare the results of a retopology with a distribution of the elements that was more rigid. Hence, less adaptation to the geometry of the object surveyed was performed, and a retopology with a final number of elements was more erratic (the control on the target number set in the software was less accurate) but with more adaptation to the geometry of the object. The high-resolution model was composed by 345,026 polygons and was simplified starting from the lower number of polygons accepted by the software, 500 polygons. Then, the number was doubled until the highest possible number in the retopology software, 95 K polygons, with a final number of nine models for the triangular and the adaptive size of 30 for retopology simplification and eight for the adaptive size of 100 for retopology, because, with this parameter, a simplification of 500 polygons did not give a proper result. All the models obtained were then compared with the high-resolution model (Table 3 and Table 4; Figure 6 and Figure 7).

The metric comparison reported in the tables shows that both the mean value and the standard deviation of the point distance between the vertexes of the reference model and the mesh of each simplified model were, as expected, higher for the retopologised model than for the triangular mesh because, with retopology, a smoothness is added to the models (The mean of the distribution gives the average position of the cloud along the normal direction, i1 and i2, and the standard deviation gives a local estimate of the point cloud roughness σ1(d) and σ2(d) along the normal direction. If outliers are expected in the data, i1 and i2 can be defined as the median of the distance distribution and the roughness is measured by the inter-quartile range. The local distance between the two clouds LM3C2(i) is then given by the distance between i1 and i2. Hence, the mean or median is the estimate of the local average position of each cloud [35], p. 8).

Given the fact that the σ of the range device is equal to 0.015 mm, the simplification aimed at creating a situation of strong difference in the final number between the different models; however, it maintained a deviation between the simplified model and high-resolution mesh not exceeding 0.2 mm, which was equal to the graphic error on paper.

Another important parameter that was considered was the number of final patches created during the automatic process of conversion from meshes to NURBS. As shown in Table 3, the number of patches in the triangular models was higher than in the other models. This can be easily explained by the fact that the patches in the NURBS had a quadrangular shape and converting a triangular mesh to NURBS involves the subdivision of each patch in two. Comparing the models derived from the two different retopology processes, it is interesting to note that the adaptive size 100 implies a number of patches slightly higher than the other process for the models with 1 K and 2 K nodes, while, by increasing the number of nodes, the result is reversed (except for the 95 K model, singularity explained with the fact that with these settings this model has a number of nodes higher than 95 K, in detail, more than 3 K nodes more than the adapt 30 model). This can be explained in a better coherence of these models on the high resolution one. The reprojection of the retopologised models on the target one permits a better geometry and a better arrangement of the elements than the patches on the surface.

The NURBS were then exported in *. step extension to create a volumetric model.

The analysis carried on was a traction analysis. The first step was to calculate the analytical result for traction on this specific object:$$\sigma =K_{t}*\sigma_{n}$$
where
$$K_{t}=1.66-theoretical\;stress\;concentration\;factor\;under\;tensile\;axial\;load\;N.$$
$$\sigma_{n}=N/A=7\mathrm{MPa}=nominal\;stress\;(=\mathrm{load}/\mathrm{cross}\;\sec\mathrm{tion}\;\mathrm{area})$$
with
A=area of the cross section (smaller diameter 7.4mm)
N=300N−applied tensile force

Thus, the analytical result is
$$\sigma =K_{t}*\sigma_{n}=1.66*7=11.62\;MPa$$

The tensile test analysis was performed on each simplified model, imposing the following parameters:-Young’s Modulus for steel 200,000 MPa;-Poisson Ratio 0.3;-Tensile load on Z axis 300 N;-Displacement as boundary conditions on the other plane face, components X = 0, Y = 0 and Z = 0;-Meshing element: 10-nodes tetrahedrons (quadratic shape functions);-Element size: from 0.1 to 2.5 mm;-Size function adaptive;-Fast transition in filling the volume.

The results are summarized in Table 5.

To better understand and read the results and to easier analyse the convergence for each model in the three simplifications processes, it was decided to convert them in percentage, giving the maximum rate of ±3% of the analytical result, as shown in Table 6, Table 7 and Table 8 and Figure 8, Figure 9 and Figure 10.

The results gave important information regarding the best solution to be adopted when using a reality-based mesh as a starting point of FEA analysis. The triangular simplified models provided the worst results regarding both the analysis itself and the convergence; the results were not homogenous, and only the 16 K model showed a real convergence, starting from the 1.5 mm volumetric mesh. The retopology method showed much better results. The models produced using the low adaptive size parameter converged from the 8 K model meshed with 1.5 mm element size to the 32 K models, meshed from 1 to 2.5 mm element size. These results are much better than the ones obtained with the triangular superficial mesh.

Finally, setting the adaptive size parameter at its higher value gave the best results. The convergence test was positive from the 2 K model meshed with 2 and 2.5 mm to the 64 K model, meshed from 0.5 to 2.5 mm element size. Compared to the other outcomes, the ones from this method were the most complete and homogeneous, even considering the results that did not go to convergence.

Another important parameter that came out from this test was that the analysis started to converge when the size of the volumetric model in the FEA software was close to the size of the superficial mesh. This information was fundamental to set the proper methodology for the tests on cultural heritage objects. For the triangular simplified models, given the inhomogeneous arrangement and dimension of the superficial elements, the comparison was conducted considering the mean value of the dimension of the elements. The convergence was reached when the models were meshed with the dimension of the tetrahedrons close to this value (Table 9, Table 10 and Table 11).

For the retopologised models, as said, the models produced with the low adaptive size parameter showed a homogeneous dimension of the elements that are perfectly square. Additionally, in this case, the convergence analysis started when the two different element sizes were almost the same (Table 10).

Finally, with the retopology method and the adaptive size parameter set at its maximum value, given the fact that especially with a lower number of nodes the shape of the elements was rectangular, the comparison was performed with the mean value. Also in this case, the convergence started when the two element sizes were similar (Table 11).

Summarizing, the models derived from a simplification with triangular superficial elements showed the worst results both in the order of processing time and, more important, structural analysis results. The two retopology methods were equivalent regarding the processing time, but when dealing with the structural results, the method adopted with the high adaptive size parameter gave the best accordance both regarding the convergence analysis and the closeness to the analytical result. This can be explained with the higher adherence of these models to the high-resolution one, even if, with a strong simplification, the models are smoother than the others. The advantage of the rectangular elements and the adaptability of the models with this process make this the best method to be used for the structural analysis of heritage directly using the reality-based meshes. This is clear looking at the highlighted values in the tables above that indicate the percentage of the convergence value of the FE analysis. The highlighted values are the ones that went to convergence; thus, the results expressed in Table 9, Table 10 and Table 11 related to different types of simplification of the mesh indicate that using a higher adaptive size parameter in the retopology process helps to reach convergence in a more distributed and coherent way.

## 5. Discussion

Before using this methodology on objects and structures that cannot be tested in the laboratory, to corroborate the results, another validation test was performed on a handmade contemporary violin, the closest to the objects to which the methodology was set for.

The process was applied to evaluate a vibroacoustic numerical model of the violin, based on accurate structural modelling [36]. The violin was surveyed with a six-axis arm laser scanner with a tolerance of 0.03 mm. The post processing started from the correction of the alignment of the different meshes acquired (Figure 11). It was not possible to perfectly correct the misalignment on one side of the model, but the mesh was cleaned and completed where the missing parts or holes were visible.

After this step, the superficial mesh was simplified using retopology with the adaptive size parameter set to 100 to obtain a 22 K model for each part of the violin. From this superficial mesh, two NURBS were created and exported in *.iges format (Figure 12a,b).

The volumetric models were then meshed in the FEA software, and some modal analyses were provided and then compared to the experimental ones (Figure 13). The results gave good accuracy compared to the laboratory ones. The slight difference between the results must be attributed to the error in the alignment of the initial mesh that caused a reduction of the thickness of one side of the violin of 0.4 mm. This test represents a solid validation of the process presented in this work because it was applied to an object with a complex geometry and for which it was possible to evaluate the results of the FE analysis with a test directly performed on the physical object. Furthermore, the modal analysis results depended on the mass and stiffness distribution along the entire model, and this evidenced the accuracy of the proposed method in respect to the actual mass and geometry of the analysed object.

After these validation tests, the methodology was applied to the statue of the Gladiator, in the Uffizi Gallery in Florence. The high-resolution model had 1,141,268 faces and was simplified to 30 K nodes. The comparison of the triangular high-resolution model and the retopologised one gave a mean of 0.0001 m and a standard deviation of 0.0004 m (Figure 14).

The simplified model was converted in NURBS, exported in *.step format and then imported in the FEA software for two different analyses: a static one imposing the gravitational load (self-weight) and a modal one, fixing 10 analyses to determine the natural mode shapes and frequencies of the object during free vibration (Figure 15). The following conditions were imposed in the structural analysis with a dimension of the volumetric element of 16 mm:-Density: 2500 kg/m^3^;-Young’s Modulus for marble 78,000 MPa;-Poisson Ratio 0.3;-Gravity on -Z axis;-Fixed support under the basement of the statue as boundary condition;-Meshing element: 10-nodes tetrahedrons;-Element size: 50 mm.

Having validated the method with the previous analyses, its application to a cultural heritage object indicates one of the most attractive applications of the method, which is to accurately determine the stress state of geometrical details (where the notch effect could case unexpected failures) with the static analysis and the global dynamic behaviour (depending upon the global schematization of the object) by means of an affordable model, both in terms of modelling and regarding the computational time.

The results show that the static analysis of the statue under its self-weight evidenced that there are not local details with severe stress concentration and that the stress is in each single point moderate and not dangerous. Further development could include occasional load due to some movement to check the proper way to move it without causing a dangerous situation.

Regarding the modal analysis, the dynamic behaviour was identified, and this can be an important aid for the evaluation of the behaviour of the statue under seismic loads, to address the proper way to protect the statue from this exceptional event. The results can be used as a term of comparison for simplified models, which respect the global mass and geometric properties of the statue, allowing a faster analysis without losing accuracy. From this point, this model can be considered itself a term of comparison for other models.

## 6. Conclusions

The use of retopology and the method proposed showed great performances using a model derived from a reality-based survey for finite element analysis. The use of the 3D measurement uncertainty as a simplification criterion allowed a considerable reduction in mesh size, maintaining a high accuracy of the simplified model compared to the high resolution one.

The main purpose of this work was benefit from the high-resolution reality-based models, considering their details, and use them for the structural analysis. The need to retain a high level of formal definition was acquired with the survey and was compatible with FEA software was the most important result.

The laboratory specimen demonstrated that the FEA results always give solutions closer to the analytical reference using volumetric models originated by the proposed method with retopology instead of a model created by a generic simplification of triangular meshes that showed not only a more arbitrary performance in the analysis but also lower accuracy.

Another important result obtained, because of the convergence analysis, was the detection of the best element size to be put in the FEA software to complete the analysis. The statement that using a size of the volumetric element close to the size of the elements of the meshes from which the volumetric model was created provided an important parameter to be set, especially when dealing with models of objects that cannot be tested in the laboratory. More information that can be acquired testing the process on a simple object and comparing the results with the analytical one more the pipeline can be robust. In this way, every single part of the process is verified, and the uncertainty of the process is minimized to the standard approximation of finite element analysis.

The tests on the violin confirmed the process on a more morphologically complex object, with the possibility to compare the results with the one performed in the laboratory.

Thanks to the tests conducted, it was also evident that the transformation of the meshes into NURBS, given the same number of point-nodes, since meshes interpolate points while NURBs approximate internal points constrained at the beginning and the end, worked better in the retopologised models, meaning a smaller number of patches. This is a fundamental point when transforming the superficial meshes into volumetric ones, both in the reduction of the processing time and, most important, in the accuracy of the results.

Applying the methodology to cultural heritage objects confirmed that it is possible to obtain an accurate FE analysis starting with an accurate simplified reality-based model, keeping the level of details and the geometric complexity of the initial high-resolution model.

The research is not closed with this work; some parts must be more deeply analysed. the first is imagining new paths of experimentation, where digital construction acquires the value of a descriptive language of exchange on several levels and is a diriment objective for critical survey and representation. Sharing and making models interoperable therefore directs the subsequent developments of the presented study. Ongoing efforts are focused on the opportunity to consider the properties of the different materials used in constructed artefacts in parallel with a more accurate segmentation analysis of the models. Having the possibility to subdivide the model in its main part (structural or decorative) and giving them the proper parameters (density, Young Modulus, Poisson’s ration, element of the mesh) gains a higher accuracy in the analysis, especially when dealing with buildings. This type of test object is more interesting considering FEA because of their structural behaviour and the higher complexity of geometry.

## Figures and Tables

**Figure 1 sensors-22-09593-f001:**
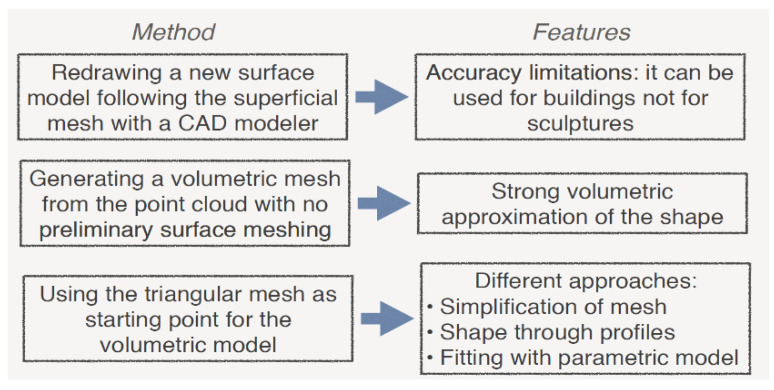
Pros and cons of the different processes used for the generation of volumetric models of cultural heritage for FEA.

**Figure 2 sensors-22-09593-f002:**
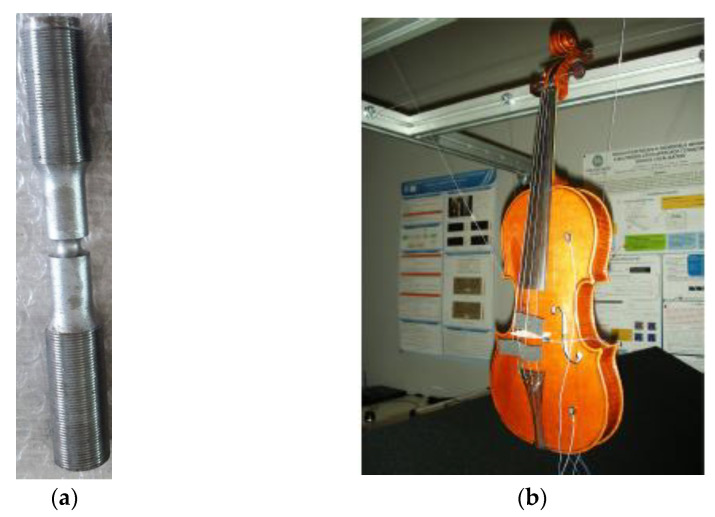
The three tested objects: (**a**) the lab specimen, (**b**) the contemporary violin, and (**c**) the statue of the Gladiator in the Uffizi Gallery.

**Figure 3 sensors-22-09593-f003:**
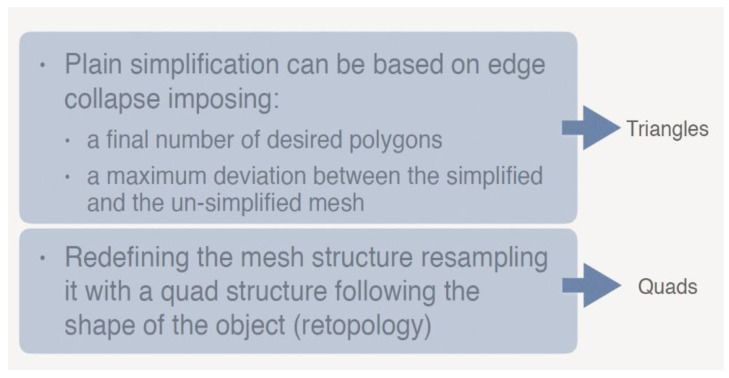
Comparison between the triangular and the retopology simplification.

**Figure 4 sensors-22-09593-f004:**
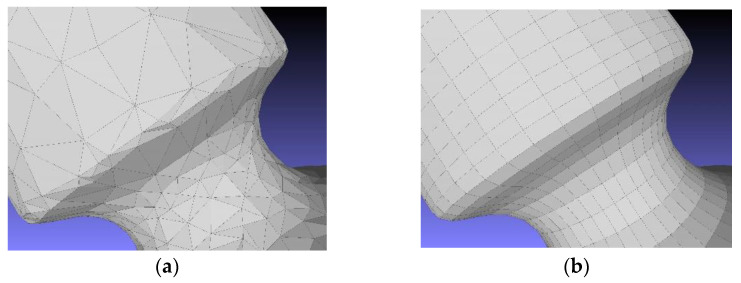
Difference between the triangular simplification (**a**) 9A and retopology simplification (**b**) on the lab specimen.

**Figure 5 sensors-22-09593-f005:**
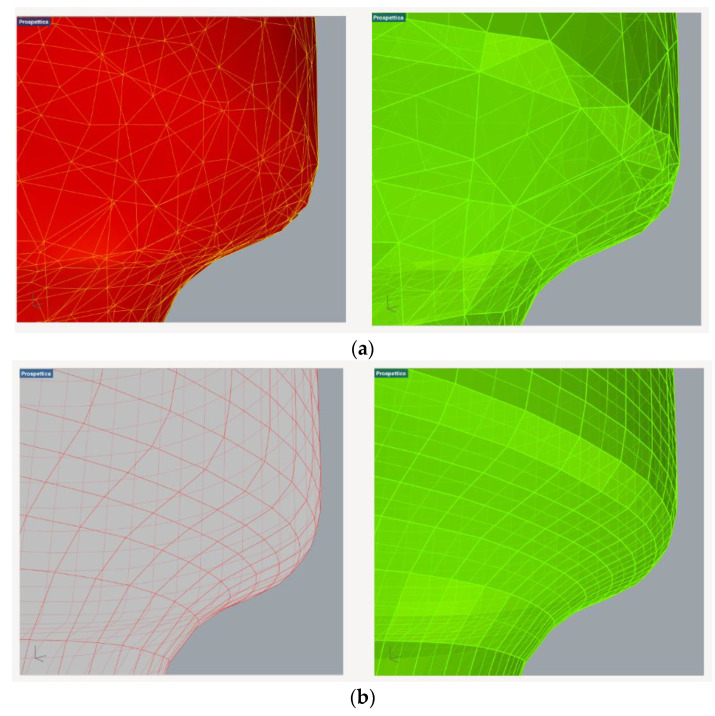
The comparison between a simplified triangular mesh and its corresponding NURBS (**a**) and the retopologised mesh with its corresponding NURBS (**b**). The meshes are represented in red (left side of the images) and the NURBS in green (right side of the images).

**Figure 6 sensors-22-09593-f006:**
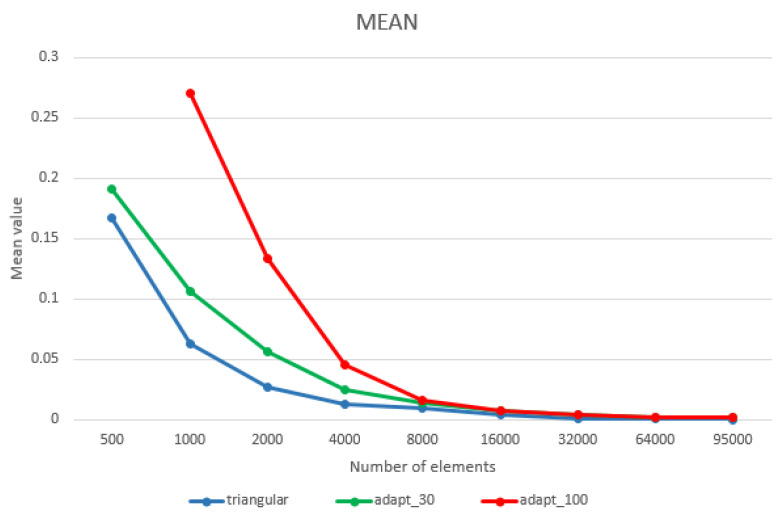
The mean in mm of the simplified models for each simplification method compared to the high-resolution one. Expressed in a graph.

**Figure 7 sensors-22-09593-f007:**
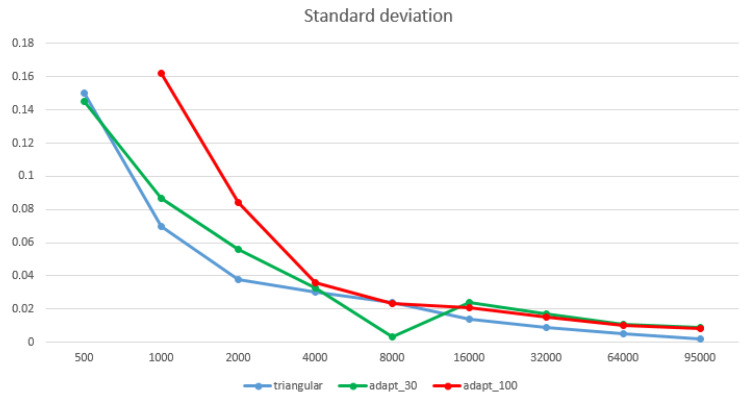
The standard deviation in mm of the simplified models for each simplification method compared to the high-resolution one. Expressed in a graph.

**Figure 8 sensors-22-09593-f008:**
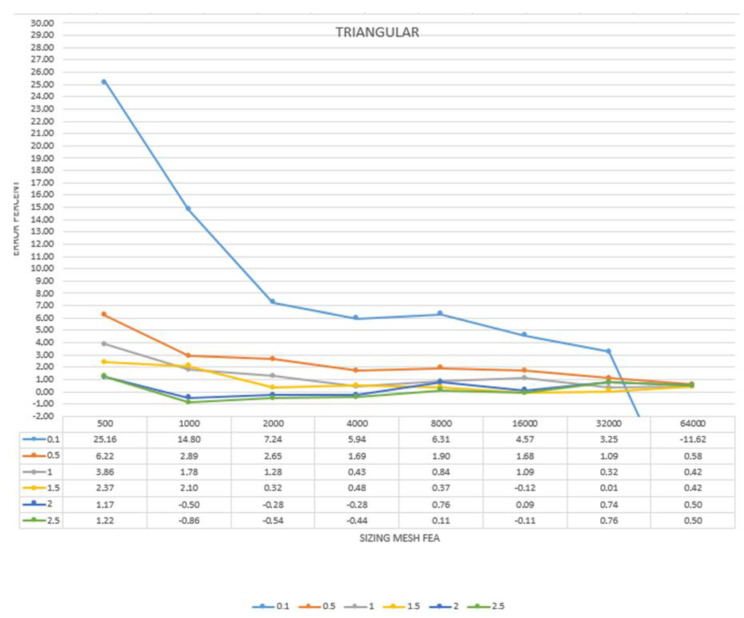
Convergence analysis for the triangular simplified models. In yellow, the results gone to convergence are expressed in percentage in relation to the result given by the analysis and the analytical result calculated for the specimen. Expressed in a graph. (Left is ERROR PERCENT).

**Figure 9 sensors-22-09593-f009:**
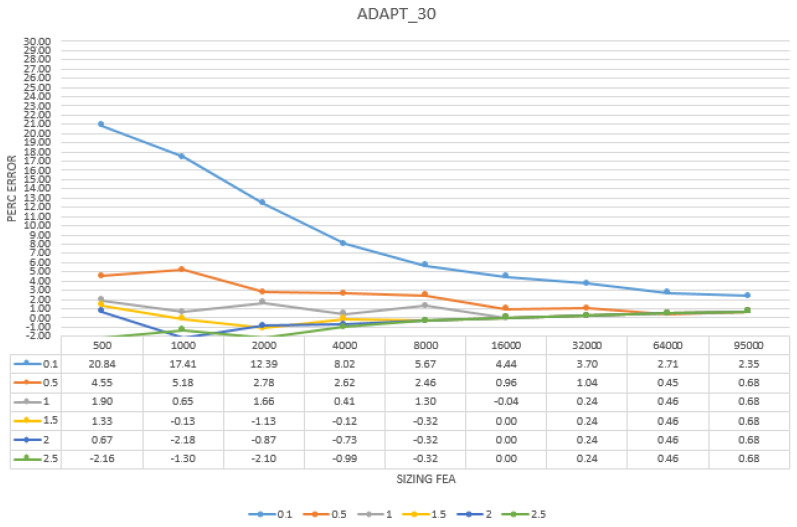
Convergence analysis for the models simplified using retopology with the adaptive size fixed at 30. Highlighted in yellow are the results on convergence expressed in percentage in relation to the result given by the analysis and the analytical result calculated for the specimen. Expressed in a graph.

**Figure 10 sensors-22-09593-f010:**
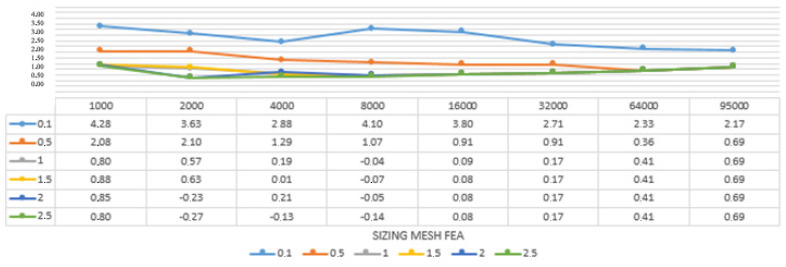
Convergence analysis for the models simplified with retopology and the adaptive size parameter fixed at 100. In yellow, the results gone to convergence are expressed in percentage in relation to the result given by the analysis and the analytical result calculated for the specimen. Expressed in a graph.

**Figure 11 sensors-22-09593-f011:**
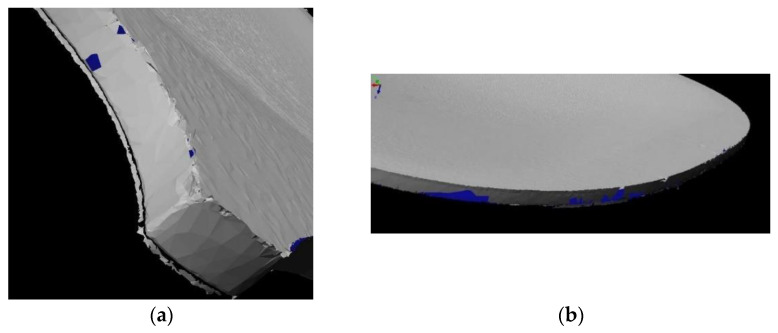
The initial mesh of the violin: (**a**) the misalignment of the different meshes on one side of the object; (**b**) holes on the surface.

**Figure 12 sensors-22-09593-f012:**
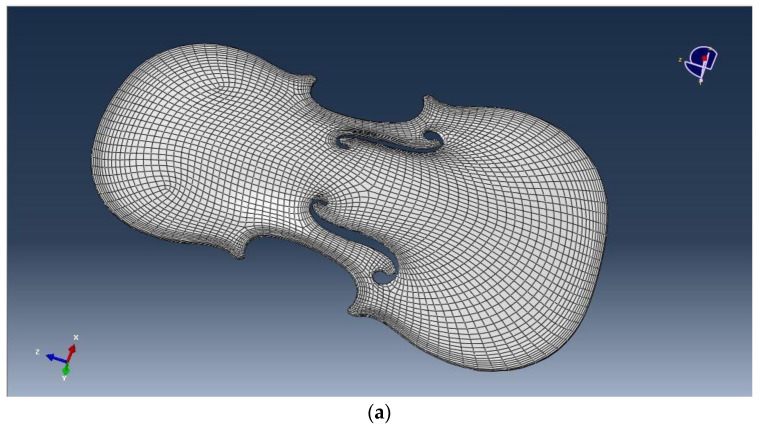
The retopologised model of the of the soundboard of the violin (**a**) and the NURBS obtained of both the soundboard and the back of the violin (**b**).

**Figure 13 sensors-22-09593-f013:**
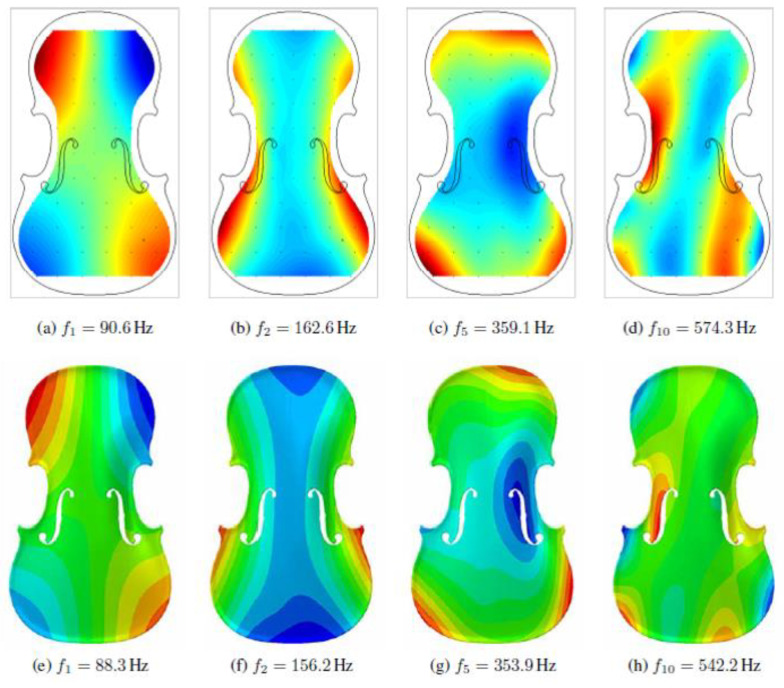
First, second, fifth and tenth vibration modes of the free soundboard from the experimental data (upper part of the image, (**a**–**d**)); same mode evaluated through the FEA (lower part of the image, (**e**–**h**)).

**Figure 14 sensors-22-09593-f014:**
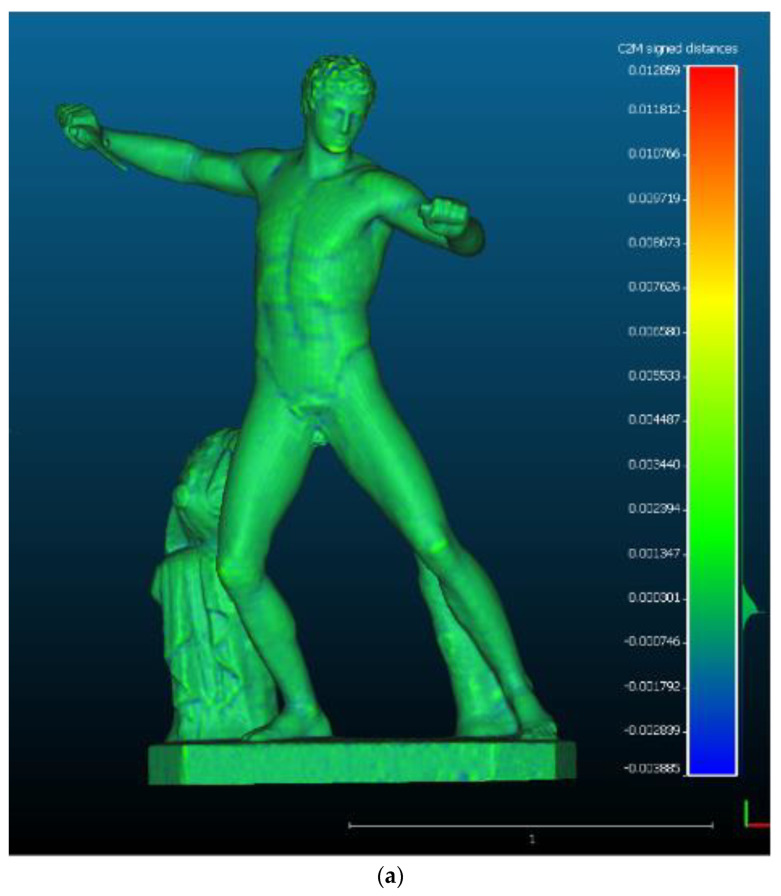
The comparison between the high resolution and the simplified retopologised model of the gladiator, where (**a**) is the graphical and visual comparison of the two meshes, and (**b**) is the gaussian distribution of the mean and the standard deviation calculated during the mesh-to-mesh comparison.

**Figure 15 sensors-22-09593-f015:**
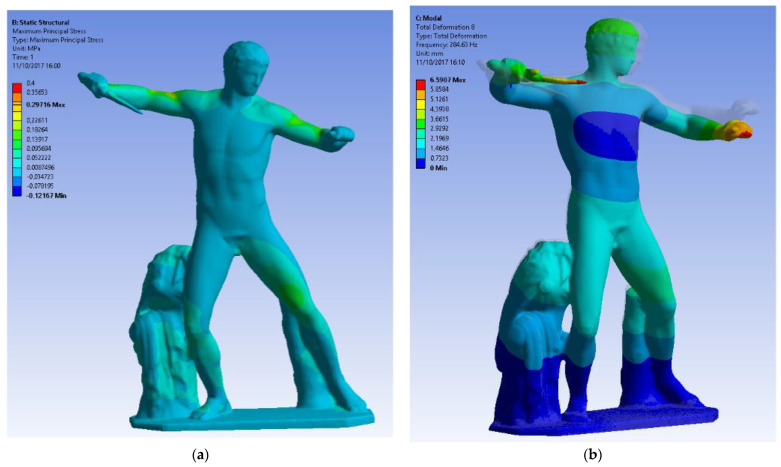
The static structural analysis imposing (**a**) the max principal stress under gravity on the –Z axis and (**b**) the modal analysis for the statue of the gladiator.

**Table 1 sensors-22-09593-t001:** Specification of the active 3D device used in the experiments.

Element	Description
Camera resolution	2.0 Mega/5.0 Mega pixel
Distance among points	0.0035~0.2 mm
Lenses	12, 25 and 50 mm
Working distance	570 mm
Principle of scan	Optical Triangulation
Dimensions	400 × 110 × 210 mm
Weight	40 N
Light	Blue LED
Unit	mm

**Table 2 sensors-22-09593-t002:** Identification of the different lenses available.

Lens	FOV/Diagonal	Distance among Points	Estimated Uncertainty
50 mm	85 (S) mm	0.044 mm	0.010 mm
25 mm	185 (S) mm	0.097 mm	0.020 mm
12 mm	370 (S) mm	0.200 mm	0.030 mm

**Table 3 sensors-22-09593-t003:** The mean in mm of the simplified models for each simplification method compared to the high-resolution one.

	500	1000	2000	4000	8000	16,000	32,000	64,000	95,000
adapt_30	0.1914	0.1063	0.0568	0.0246	0.0138	0.0078	0.0043	0.0024	0.0019
adapt_100		0.2707	0.1337	0.0457	0.0164	0.0078	0.0038	0.0022	0.0017
triangular	0.1678	0.0631	0.0269	0.0131	0.0091	0.0037	0.0011	0.0005	0.0003

**Table 4 sensors-22-09593-t004:** The standard deviation in mm of the simplified models for each simplification method compared to the high-resolution one.

	500	1000	2000	4000	8000	16,000	32,000	64,000	95,000
adapt_30	0.145	0.087	0.056	0.033	0.003	0.024	0.017	0.011	0.009
adapt_100		0.162	0.084	0.036	0.023	0.021	0.015	0.01	0.008
triangular	0.15	0.07	0.038	0.03	0.024	0.014	0.009	0.005	0.002

**Table 5 sensors-22-09593-t005:** The results expressed in MPa of the FEA analysis on traction on the different models of the laboratory specimen. The different colours highlight the values that were recurrent in the analysis even with different element dimensions.

0.1 mm size mesh Ansys									
	500	1000	2000	4000	8000	16,000	32,000	64,000	95,000
triangular	36,784	26,421	18,862	17,555	17,927	16,192	14,874	/	/
adapt_30	32,464	29,033	24,009	19,641	17,293	16,056	15,317	14,328	13,974
adapt_100		15,895	15,252	14.5	15,715	15,424	14,583	13,945	13,785
0.5 mm size mesh Ansys									
	500	1000	2000	4000	8000	16,000	32,000	64,000	95,000
triangular	17,835	14,506	14,269	13,312	13,521	13,303	12,711	12,201	/
adapt_30	16,171	16,803	14,395	14,243	14,079	12,584	12.66	12,067	12,304
adapt_100	/	13,704	13,718	12,912	12,686	12,525	12,527	11,983	12,305
1 mm size mesh Ansys									
	500	1000	2000	4000	8000	16,000	32,000	64,000	95,000
triangular	15,478	13,401	12.9	12.05	12,459	12,707	* 11,943 *	*12.04*	/
adapt_30	13,522	12.27	13,282	12,034	12,916	11,584	11,855	12,084	12,304
adapt_100	/	12,415	12,187	11,809	15,575	11,714	11,792	12,026	12,305
1.5 mm size mesh Ansys									
	500	1000	2000	4000	8000	16,000	32,000	64,000	95,000
triangular	13,985	13,722	11,935	12,102	11,986	*11,499*	* 11,634 *	* 12,037 *	/
adapt_30	12,954	11,493	10,486	11,502	11,299	11,616	11,855	12,084	12,304
adapt_100	/	12,503	12,253	11.63	11,546	11,695	11,792	12,026	12,305
2 mm size mesh Ansys									
	500	1000	2000	4000	8000	16,000	32,000	64,000	95,000
triangular	12,792	11,119	11,338	11,388	12,375	*11,714*	*12,358*	*12.12*	/
adapt_30	12,286	9,4354	10.75	10,885	11,299	11,616	11,855	12,084	12,304
adapt_100	/	12,467	11,385	11,826	11,567	11,695	11,792	12,026	21,305
2.5 mm size mesh Ansys									
	500	1000	2000	4000	8000	16,000	32,000	64,000	95,000
triangular	12,838	10,759	11.08	11.18	**11,734**	** *11,505* **	** *12,383* **	** *12.12* **	**/**
adapt_30	94,614	10,324	95,181	10,629	11,299	11,616	11,855	12,084	12,304
adapt_100	/	** 12,418 **	**11,347**	**11,489**	**11,479**	11,695	11,792	12,026	12,305

**Table 6 sensors-22-09593-t006:** Convergence analysis for the triangular simplified models. In yellow, the results gone to convergence are expressed in percentage in relation to the result given by the analysis and the analytical result calculated for the specimen.

	**500**	**1000**	**2000**	**4000**	**8000**	**16,000**	**32,000**	**64,000**
**0.1**	2.17	1.27	0.62	0.51	0.54	0.39	0.28	−1.00
**0**	0.53	0.25	0.23	0.15	0.16	0.14	0.09	0.05
**1**	0.33	0.15	0.11	0.04	0.07	0.09	0.03	0.04
**1.5**	0.20	0.18	0.03	0.04	0.03	−0.01	0.00	0.04
**2**	0.10	−0.04	−0.02	−0.02	0.06	0.01	0.06	0.04
**2.5**	0.10	−0.07	−0.05	−0.04	0.01	−0.01	0.07	0.04

**Table 7 sensors-22-09593-t007:** Convergence analysis for the models simplified using retopology with the adaptive size fixed at 30. Highlighted in yellow are the results on convergence expressed in percentage in relation to the result given by the analysis and the analytical result calculated for the specimen.

	**500**	**1000**	**2000**	**4000**	**8000**	**16,000**	**32,000**	**64,000**	**95,000**
**0.1**	1.79	1.50	1.07	0.69	0.49	0.38	0.32	0.23	0.20
**0.5**	0.39	0.45	0.24	0.23	0.21	0.08	0.09	0.04	0.06
**1**	0.16	0.06	0.14	0.04	0.11	0.00	0.02	0.04	0.06
**1.5**	0.11	−0.01	−0.10	−0.01	−0.03	0.00	0.02	0.04	0.06
**2**	0.06	−0.19	−0.07	−0.06	−0.03	0.00	0.02	0.04	0.06
**2.5**	−0.19	−0.11	−0.18	−0.09	−0.03	0.00	0.02	0.04	0.06

**Table 8 sensors-22-09593-t008:** Convergence analysis for the models simplified with retopology and the adaptive size parameter fixed at 100. In yellow, the results gone to convergence are expressed in percentage in relation to the result given by the analysis and the analytical result calculated for the specimen.

	**1000**	**2000**	**4000**	**8000**	**16,000**	**32,000**	**64,000**	**95,000**
**0.1**	0.37	0.31	0.25	0.35	0.33	0.23	0.20	0.19
**0.5**	0.18	0.18	0.11	0.09	0.08	0.08	0.03	0.06
**1**	0.07	0.05	0.02	0.00	0.01	0.01	0.03	0.06
**1.5**	0.08	0.05	0.00	−0.01	0.01	0.01	0.03	0.06
**2**	0.07	−0.02	0.02	0.00	0.01	0.01	0.03	0.06
**2.5**	0.07	−0.02	−0.01	−0.01	0.01	0.01	0.03	0.06

**Table 9 sensors-22-09593-t009:** The comparison between the dimension of the element of the superficial meshes and the volumetric element in FEA for the triangular simplified models. The column represents the dimension of the superficial mesh element, and the rows represent the dimension of the volumetric elements in the FEA software.

**Element Dim**	**3–10**	**1.3–6**	**1–5**	**0.6–4**	**0.4–3**	**0.4–2**	**0.3–1.2**	**0.2–1.2**
**0.1**	2.17	1.27	0.62	0.51	0.54	0.39	0.28	−1.00
**0.5**	0.53	0.25	0.23	0.15	0.16	0.14	0.09	0.05
**1**	0.33	0.15	0.11	0.04	0.07	0.09	0.03	0.04
**1.5**	0.20	0.18	0.03	0.04	0.03	−0.01	0.00	0.04
**2**	0.10	−0.04	−0.02	−0.02	0.06	0.01	0.06	0.04
**2.5**	0.10	−0.07	−0.05	−0.04	0.01	−0.01	0.07	0.04

**Table 10 sensors-22-09593-t010:** The comparison between the dimension of the superficial and the volumetric element for the retopologised models with the adaptive size parameter set to 30. The columns represent the size of the quadrangular mesh elements, while the rows represent the size of the volumetric ones.

**Element Dim**	**5**	**3.7**	**2.7**	**1.8**	**1.2**	**0.9**	**0.6**	**0.4**	**0.3**
**0.1**	1.79	1.50	1.07	0.69	0.49	0.38	0.32	0.23	0.20
**0.5**	0.39	0.45	0.24	0.23	0.21	0.08	0.09	0.04	0.06
**1**	0.16	0.06	0.14	0.04	0.11	0.00	0.02	0.04	0.06
**1.5**	0.11	−0.01	−0.10	−0.01	−0.03	0.00	0.02	0.04	0.06
**2**	0.06	−0.19	−0.07	−0.06	−0.03	0.00	0.02	0.04	0.06
**2.5**	−0.19	−0.11	−0.18	−0.09	−0.03	0.00	0.02	0.04	0.06

**Table 11 sensors-22-09593-t011:** The comparison between the two element sizes in the retopologised models with an adaptive size parameter of 100. The columns represent the size of the superficial element, and the rows represent sizes of the volumetric ones.

**Element Dim**	**6 × 1.8**	**4 × 1.8**	**2.5 × 1.7**	**1.4**	**1 × 0.8**	**0.7**	**0.4**	**0.3**
**0.1**	0.37	0.31	0.25	0.35	0.33	0.23	0.20	0.19
**0.5**	0.18	0.18	0.11	0.09	0.08	0.08	0.03	0.06
**1**	0.07	0.05	0.02	0.00	0.01	0.01	0.03	0.06
**1.5**	0.08	0.05	0.00	−0.01	0.01	0.01	0.03	0.06
**2**	0.07	−0.02	0.02	0.00	0.01	0.01	0.03	0.06
**2.5**	0.07	−0.02	−0.01	−0.01	0.01	0.01	0.03	0.06

## Data Availability

Not applicable.

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
