# Peer review of "3D Reality-Based Survey and Retopology for Structural Analysis of Cultural Heritage"

_sensors, 2022, doi:10.3390/s22249593_

Round 1
Reviewer 1 Report
First of all, I thank the authors for an exciting article. Their research is genuinely multidisciplinary, with a broad reach and many practical applications. I apprehend the article in two ways:
- introduction of a new technology of object transformation into the FEA environment, and
- applications of this technology to specific solutions (Chapter 4).
I have several formal and professional comments and questions about the article. I'll start with the formal ones:
- The abbreviation FEA (line 20) appears before its definition.
- the division of the images is weird (e.g., Fig. 2 into parts a, b, and c, - I suggest changing the composition and description; the same applies to Fig. 5, 6, 7, 9, 10; in my opinion, Fig. 8 is marked correctly).
- I suggest dividing the tables merged with images (e.g., Tab. 4, etc.).
- In most tables, it is not clear what physical entity is captured in them, and the units are also missing - this is necessary to state nearby or in the tables themselves.
- Typo in Tab. 1: Kg
- Capitalized K is often used in the text - is this correct?
- Table captions should be above the tables and not below them.
- I am not convinced of the correctness of using the comma as a decimal point. It is possible that you need to use a dot instead - I recommend checking.
- The Y-axes of some graphs are "compressed" with scale descriptions (e.g., Fig. 7 and 8).
Regarding professional comments/questions, I have the following:
- Line 74: You use the so-called "surface" model. The Boundary Element Method (BEM) is a typical and natural environment for a task constructed in this way. Why do you use FEM? What is the advantage of FEM over BEM for the given conditions?
- What is the sensor size for the camera used? Is this an APS-C sensor? Furthermore, what are the lenses' parameters, especially regarding the barrel effect, pincushion effect, etc.? What impact can these flaws have on the final scanning?
- Line 244: have you considered using higher-order elements in FEA?
- Chapter 2.4: do you describe your own analysis conditions or FEM in general? As long as the chapter is devoted to a general description of FEM, I disagree with a lot of information presented here.
- Line 354: what are the criteria for determining accuracy? Do you know the exact result of the analysis in advance?
- Line 374: "accurate solution in FEA" is a mystification. FEA is inherently inaccurate.
- Lines 373 – 376: The content of this sentence cannot be considered generally correct.
- Line 386: "size parameter at 30" - please explain what entity and in which units it is about here.
- Tab. 4: What should I understand by "Mean" and "Standard deviation," and in what units?
- Tab. 4: What is the explanation for the point: adapt_30/8000/0.003?
- Line 402: where should I look for the description of the "high-resolution" method in the text (specifically, which one is it?)?
- Table 5: Recurrent values are highlighted. Why are these important?
- Lines 513 – 514: "The advantage of the rectangular elements and the more adaptability of the models with this process" - it depends on what character of the resulting quantity is expected. When meshing as described, 3D is essentially reduced to 2D. In this case, triangular elements, in principle, lead to a linear distribution of the resulting quantity over the given element. In the case of rectangular elements, it is not possible to preserve the linearity in 2D, and the system handles non-linear functions. Here arises the question of the expected distribution of the resulting quantity and the agreement with the shape of the used weighting functions (it is likely a polynomial one).
Despite my numerous comments, I understand the whole article very positively, and I thank the authors in advance for their answers.
Author Response
First of all, I thank the authors for an exciting article. Their research is genuinely multidisciplinary, with a broad reach and many practical applications. I apprehend the article in two ways:
- introduction of a new technology of object transformation into the FEA environment, and
- applications of this technology to specific solutions (Chapter 4).
The authors want to thank very much the reviewer for the clear and valuable indications and the revision done.
I have several formal and professional comments and questions about the article. I'll start with the formal ones:
- The abbreviation FEA (line 20) appears before its definition.
Fixed
- the division of the images is weird (e.g., Fig. 2 into parts a, b, and c, - I suggest changing the composition and description; the same applies to Fig. 5, 6, 7, 9, 10; in my opinion, Fig. 8 is marked correctly).
Fixed
- I suggest dividing the tables merged with images (e.g., Tab. 4, etc.).
Fixed
- In most tables, it is not clear what physical entity is captured in them, and the units are also missing - this is necessary to state nearby or in the tables themselves.
- Typo in Tab. 1: Kg
- Capitalized K is often used in the text - is this correct?
Fixed
- Table captions should be above the tables and not below them.
Fixed
- I am not convinced of the correctness of using the comma as a decimal point. It is possible that you need to use a dot instead - I recommend checking.
Used point
- The Y-axes of some graphs are "compressed" with scale descriptions (e.g., Fig. 7 and 8).
Fixed. The images are bigger. Unfortunately, the values are compressed because the program took all the data error in the simulation even if the graph is related to few data. We hope is clearer now.
Regarding professional comments/questions, I have the following:
- Line 74: You use the so-called "surface" model. The Boundary Element Method (BEM) is a typical and natural environment for a task constructed in this way. Why do you use FEM? What is the advantage of FEM over BEM for the given conditions?
Thanks for your pertinent comments. Indeed, it is true that the BEM is attractive for this application since we need to model just the surface, that is the output of the measurement. However, the BEM is not widely used in practical applications and there are few BEM codes for structural analysis commercially available, mainly oriented toward fracture mechanics analysis. In view of wide spreading the possible applications of the proposed method and making easier its implementation in possible applications we think that FEM is the right tool.
- What is the sensor size for the camera used? Is this an APS-C sensor? Furthermore, what are the lenses' parameters, especially regarding the barrel effect, pincushion effect, etc.? What impact can these flaws have on the final scanning?
Fixed in the text
- Line 244: have you considered using higher-order elements in FEA?
We have not used higher order elements because the output of re-topology is a mesh with nodes at the end point of each side (at the corners). The implementation of higher order elements needs the definition of mid-side nodes, and this makes less immediate and application of the method. However, we checked the accuracy of the solution considering the lab sample, with a known analytical solution. In this way we were able to correctly choose the right mesh density with linear elements.
- Chapter 2.4: do you describe your own analysis conditions or FEM in general? As long as the chapter is devoted to a general description of FEM, I disagree with a lot of information presented here.
Fixed in the text
- Line 354: what are the criteria for determining accuracy? Do you know the exact result of the analysis in advance?
We know the exact result for the lab sample, while for the violin we compared the results with experimental measurements. The statue is the case study, and the solution is not known.
- Line 374: "accurate solution in FEA" is a mystification. FEA is inherently inaccurate.
When we consider accuracy, we are considering solutions that can be used for practical applications without the risk that they can lead to wrong assessment and evaluation, which could be quite dangerous. So, it is true that FE is always approximate and never exact, but at the same time, we have criteria and tools to estimate if a solution is enough accurate or must be refined. Therefore, we have done a convergence analysis and we have considered cases where we have the analytical solutions or experimental results to validate the proposed method.
- Lines 373 – 376: The content of this sentence cannot be considered generally correct.
Probably the sentence was confused. We rephrased it and hoped it can be now clearly interpreted.
- Line 386: "size parameter at 30" - please explain what entity and in which units it is about here.
- Tab. 4: What should I understand by "Mean" and "Standard deviation," and in what units?
The units are mm as indicated in the reference of the table in the text.
- Tab. 4: What is the explanation for the point: adapt_30/8000/0.003?
- Line 402: where should I look for the description of the "high-resolution" method in the text (specifically, which one is it?)?
- Table 5: Recurrent values are highlighted. Why are these important?
Fixed in the text
- Lines 513 – 514: "The advantage of the rectangular elements and the more adaptability of the models with this process" - it depends on what character of the resulting quantity is expected. When meshing as described, 3D is essentially reduced to 2D. In this case, triangular elements, in principle, lead to a linear distribution of the resulting quantity over the given element. In the case of rectangular elements, it is not possible to preserve the linearity in 2D, and the system handles non-linear functions. Here arises the question of the expected distribution of the resulting quantity and the agreement with the shape of the used weighting functions (it is likely a polynomial one).
Referring to 2D analysis, both triangular and rectangular elements with 3 and 4 nodes respectively, have linear shape functions. The difference is the number of terms of the linear polynomial describing the geometry and the displacement field. The difference is that triangular elements have a constant stress/strain distribution inside the element while in rectangular elements the trend is linear.
Another reason why rectangular elements are generally preferred is that the triangular elements overestimate the real stiffness of the material, thus underestimating the resulting stress.
Similar concepts can be drawn is 3D considering tetrahedral elements and hexahedral elements without mid-side nodes.
So, we do not see a problem in terms of preservation of the linearity for the reasons just mentioned.
Reviewer 2 Report
Some doubts about the scan of the heritage objects, maybe heritage buildings can be analysed as novelty research objects.
Conclusions can be improved and based on the numerical results.
Author Response
Some doubts about the scan of the heritage objects, maybe heritage buildings can be analysed as novelty research objects.
Conclusions can be improved and based on the numerical results.
Thank you very much for your note. We added few sentences in the text.
Round 2
Reviewer 1 Report
Dear Authors,
All essential questions/comments have been commented on and answered. I do not always agree with the authors' response; nevertheless, I consider the article suitable for publication.
However, there are still a few formal matters that need to be resolved - the tables are indeed detached from the figures, but the figure caption is missing.
The technical quality of the images must be improved - e.g., the Y-axe scale descriptions (graph on page 18) - I am sure this technical problem has a solution.
Author Response
Thank you very much.
We have added number of figures and fixed the image at èpage 18.
Best Regards.